# Spectroscopic and Structural Study of Some Oligosilanylalkyne Complexes of Cobalt, Molybdenum and Nickel

**DOI:** 10.3390/molecules24010205

**Published:** 2019-01-08

**Authors:** Michaela Zirngast, Christoph Marschner, Judith Baumgartner

**Affiliations:** Institut für Anorganische Chemie, Technische Universität Graz, Stremayrgasse 9, A-8010 Graz, Austria; michaela.zirngast@allnex.com

**Keywords:** oligosilanylalkynes, transition metal alkyne complexes, Si-Si bond activation

## Abstract

Metal induced stabilization of α-carbocations is well known for cobalt- and molybdenum complexed propargyl cations. The same principle also allows access to reactivity enhancement of metal coordinated halo- and hydrosilylalkynes. In a previous study, we have shown that coordination of oligosilanylalkynes to the dicobalthexacarbonyl fragment induces striking reactivity to the oligosilanyl part. The current paper extends this set of oligosilanylalkyne complexes to a number of new dicobalthexacarbonyl complexes but also to 1,2-bis(cyclopentadienyl)tetracarbonyldimolybdenum and (dippe)Ni complexes. NMR-Spectroscopic and crystallographic analysis of the obtained complexes clearly show that the dimetallic cobalt and molybdenum complexes cause rehybridization of the alkyne carbon atoms to sp^3^, while in the nickel complexes one π-bond of the alkyne is retained. For the dicobalt and dimolybdenum complexes, strongly deshielded ^29^Si NMR resonances of the attached silicon atoms indicate enhanced reactivity, whereas the ^29^Si NMR shifts of the respective nickel complexes are similar to that of respective vinylsilanes.

## 1. Introduction

It is well known that the reactivity of organic molecules can be altered dramatically by coordination to a metal fragment. Alkene coordination to Pd(II) represents a typical example, where electron density from the π-bond is shifted toward the metal. The thus caused polarization allows nucleophilic attack on the olefin, which is otherwise fairly unreactive [1]. 

Other particularly interesting cases of reactivity enhancement by metal coordination include metal induced stabilization of α-carbocations such as observed for dicobalt- and dimolybdenum complexed propargyl cations. The Nicholas reaction, which is the reaction of a dicobalt-coordinated propargyl cation with a nucleophile, depends on this effect [2,3,4,5]. As also silyl-substituted alkynes can be subjected to complexation with dicobaltoctacarbonyl [6,7,8,9], the question arose whether some kind of *sila*-Nicholas reaction might be possible. Corriu and co-workers have indeed found that dicobalt complexed methoxy- and halo-silylalkynes undergo facile alkylation, reduction, hydrolysis, and halogenation reactions [10]. In a subsequent study, enhanced reactivity of the Si-H bond of a coordinated hydrosilylalkyne was demonstrated and further investigated [11]. Whether the coordination of alkyne is essential for increased reactivity of hydrosilylalkynes could not be established as it was found that also non-coordinated hydrosilylalkynes would react with alcohols in the presence of 10 mol% of a Co_2_(CO)_6_-tolane complex [11]. 

For dicobalthexacarbonyl stabilized propargyl cations a fair number of isolated and well characterized examples are known [12,13]. For the silicon case, silylium ions are likely to be involved in reactions of the respective complexes, as was demonstrated by Corriu and co-workers, who reacted their dicobalthexacarbonyl coordinated hydrosilylalkyne with Ph_3_C[BF_4_] to obtain the respective dicobalthexacarbonyl complexed fluorosilane [11]. Theoretical and experimental aspects of the formation of transition metal-stabilized silylium ions were further studied by Brook and co-workers using again dicobalt- but also dimolybdenum-complexes for the coordination of silylated alkynes [14,15,16]. 

In order to check, whether complexation with dicobaltoctacarbonyl can activate even Si-Si bonds, our group prepared a number of dicobalthexacarbonyl coordinated tris(trimethylsilyl)silylalkyne complexes [17] (see below). These proved to be very reactive against a number of reagents. To provide a simplified system, we thus prepared a pentamethyldisilanylethyne dicobalthexacarbonyl complex, which underwent clean reactions with MeOH and H_2_O to the respective methoxysilane and disiloxane complexes (Scheme 1) [17]. 

NMR spectroscopic analysis of the obtained oligosilylalkyne dicobalt complexes showed the respective α-silicon atoms to be substantially deshielded, which is in line with the observed enhanced electrophilicity [17]. The current study aims at extending spectroscopic and structural studies to more elaborate oligosilanyl alkynes and to other transition metals.

## 2. Results and Discussion

To gain more detailed insight into the spectroscopic and structural properties of metal complexed oligosilanylated alkynes we decided to compare the properties of the already studied examples of cobalt complexes together with a couple of additional alkyne cobalt complexes to a few related complexes of molybdenum and nickel. 

### 2.1. Cobalt Complexes

As depicted in Scheme 2 a number of a oligosilanylalkynes (**1**–**6**) were reacted previously with Co_2_(CO)_8_ to give the expected dicobalthexacarbonyl complexes **1a**–**6a** [17] (Scheme 2). Molecular structures of several of these compounds were studied by single crystal XRD analysis (Table 2). Table 1 features ^13^C (alkyne C atoms) and ^29^Si (alkynyl attached Si atoms) NMR chemical shifts of the employed free alkynes together with the respective resonances for the metal coordinated compounds. 

In addition to complexes **1a**–**6a**, we decided to convert also the known alkynes **7** [18], **8** [19], **9** [20], and **10** [18], as well as the novel compound **11** to dicobalthexacarbonyl complexes. Compounds **7** and **11** are similar to tris(trimethylsilyl)silylphenylacetylene (**1**), but **7** contains a tris(trimethylsilyl)germyl group instead of the tris(trimethylsilyl)silyl unit and **11** features a 2-thienyl substituent instead of the phenyl group. As expected, both alkynes underwent smooth reaction to the respective dicobalthexacarbonyl complexes **7a** and **11a** (Scheme 3).

Previously, we noted that (Me_3_Si)_3_SiC≡CSiMe_3_ and even (Me_3_Si)_3_SiC≡CMe do not react with Co_2_(CO)_8_ [17]. Since the respective dicobalthexacarbonyl complex of Me_3_SiC≡CSiMe_3_ is known [6,9], we reasoned that the presence of the rather bulky (Me_3_Si)_3_Si group requires a small (H) or flat (Ph) substituent on the other side of the ethynyl group in order to allow effective approach of Co_2_(CO)_8_ to the C≡C unit. Since the steric properties of the Me_3_SiSiMe_2_ substituent are significantly less demanding compared to (Me_3_Si)_3_Si, we assumed that the presence of another small silyl group on the alkyne would nevertheless permit coordination to the dicobalthexacarbonyl fragment. For that reason, we prepared Me_5_Si_2_C≡CSi_2_Me_5_ (**8**) and Me_5_Si_2_C≡CSiMe_3_ (**9**) containing either two or one pentamethyldisilanyl substituents. Both compounds smoothly reacted with Co_2_(CO)_8_ to give complexes **8a** and **9a** (Scheme 3).

Substrate **10** is a butadiyne with two terminal tris(trimethylsilyl)silyl groups. Although we were able to achieve complexation of two alkyne units with Co_2_(CO)_8_ in 1,4-bis[tris(trimethylsilyl)silylethynyl]benzene and dimethyldi(phenylethynyl)silane [17], compound **10** permits only metalation of one triple bond, leading to formation of complex **10a** with tris(trimethylsilyl)silyl and tris(trimethylsilyl)silylethynyl substituents (Scheme 3). Metalation of both triple bonds of 1,4-bis(trimethylsilyl)butyne with 2 equiv. Co_2_(CO)_8_ is known [8,21] but again it seems that the steric bulk of the tris(trimethylsilyl)silyl group prevents this process for **10**.

With the intention to introduce further metal coordination sites, we decided to study replacing phenyl substituents at the alkyne by thienyl groups. The reaction of the thus obtained 2,4 bis[tris(trimethylsilyl)silylethynyl]thiophene (**12**) with 2 equiv. Co_2_(CO)_8_ can be considered as a variation of the previously reported reaction of 1,4-bis[tris(trimethylsilyl)silylethynyl]benzene [17]. As expected, both alkyne parts of **12** underwent complexation with Co_2_(CO)_6_ fragments to give complex **12a** (Scheme 4). 

### 2.2. Molybdenum Complexes

Apart from the dicobalthexacarbonyl unit, also other metal fragments are known to stabilize α-carbocations. The Cp_2_Mo_2_(CO)_4_ unit is one of these and therefore we were interested to utilize some of our silylalkynes for the preparation of dimolybdenum complexes. Thermal dissociation of 2 CO from Cp_2_Mo_2_(CO)_6_ delivers Cp_2_Mo_2_(CO)_4_ [22], which is known to add easily to alkynes [23]. For reasons of comparison, we subjected silylated alkynes **2** and **6** to metalation conditions with Cp_2_Mo_2_(CO)_4_ and obtained complexes **2b** and **6b** in acceptable yields (Scheme 5). 

### 2.3. Nickel Complexes

To study also mononuclear complexes of late metals, we decided to prepare Ni(0) complexes of a number of oligosilanylated alkynes. As the nickel fragment of choice (dippe)Ni (dippe = 1,2-di-*iso*-propylphosphinoethylene) was selected. Formation of the desired complexes (**2c**, **3c**, **6c**, and **13c**) was achieved by reacting four different silylated alkynes (**2**, **3**, **6**, and **13**) with (COD)Ni(dippe) [24] in toluene at 65 °C (Scheme 6). 

Related Ni(0) complexes of silylated alkynes have been studied before [25,26,27,28,29,30,31,32,33] mainly for spectroscopic analysis but also with respect to bond activating properties [27,34]. 

### 2.4. NMR Spectroscopic Analysis

A substantial number of dicobalthexacarbonyl alkynyl complexes has been studied using NMR spectroscopy, with particular emphasis on ^13^C NMR properties of the coordinated alkyne carbon atoms [9,35,36]. It is interesting to note that the differences in the chemical shifts of free and coordinated alkyne atoms are surprisingly small. Nevertheless, in accordance also with structural information derived from crystal structure analytical studies, the coordination to the dicobalthexacarbonyl fragment goes along with a change in hybridization of the involved carbon atoms from sp to sp^3^.

Previously, we have shown that NMR spectroscopic analysis of the complexed oligosilanylalkynes provides a good tool to judge the degree of associated Si-Si bond activation. In particular, the ^29^Si NMR chemical shift values of the silicon atoms attached to the alkyne unit proved to be significant [17]. Table 1 details ^13^C and ^29^Si NMR shifts of the alkyne carbons and the attached silicon atoms of the free alkynes and the respective metal complexes. 

The difference in electron donating abilities of phenyl and 2-thienyl units is clearly reflected by the alkynyl ^13^C NMR resonances of 1-(tris(trimethylsilyl)silyl)-2-(thien-2′-yl)ethyne **11** and 1-(tris(trimethylsilyl)silyl)-2-phenylethyne **1** (Table 1). Compared to the values of **1**, the thienyl substituted carbon (δ = 100.4 ppm) of **11** is shifted some 8 ppm down-field, while the respective ß-carbon (δ = 93.5 ppm) experiences some 5 ppm shift in the opposite direction. The ^29^Si NMR shift of the attached tris(trimethylsilyl)silyl group is more or less identical for **1** and **11**. The alkynyl ^13^C NMR resonances of butadiyne **10** (δ = 79.2 (Si) and 93.4 (C) ppm) are similar to that of tris(trimethylsilyl)silylethyne (**2**), reflecting not much substituent influence on the not silylated side of the alkyne. The disilylated alkynes **8** and **9** show that the effect on the α-carbon is quite similar (ca. +17.5 ppm) for SiMe_3_ and Si_2_Me_5_ but that Si_2_Me_5_ exhibits a stronger effect on the β-carbon than SiMe_3_. Again the ^29^Si NMR shifts of alkynyl substituted SiMe_3_ (δ = ca. −18.5 ppm) and Si_2_Me_5_ (δ = ca. −38 ppm) are not much affected by the other substituent on the alkyne. Assignment and approximate prediction of alkynyl ^13^C shifts is conveniently accomplished by using an increment system as outlined earlier by us [37].

In accordance with the fact that **1** and **11** feature remarkably different alkyne ^13^C resonances also their coordination behavior to the Co_2_(CO)_6_ unit is different. While both silicon substituted carbon atoms experience an up-field shift (see Table 1), the aryl substituted atom is shifted to lower field for the case of **1** being converted to **1a**, whereas the thienyl substituted carbon of complex **11a** is shifted about 4 ppm to higher field. Again the central ^29^Si resonances of the Si(SiMe_3_)_3_ groups are practically identical for **1a** and **11a** (Table 1).

For the disubstituted alkynes **8** (δ = 115.3 ppm) and **9** (δ = 112.9 and 116.5 ppm) the ^13^C values of the alkyne carbon atoms are similar to what is known for bis(trimethylsilyl)acetylene (BTMSA) (δ = 113 ppm) [9]. Also, the effect of coordination to the Co_2_(CO)_6_ fragment is mostly identical leading to ^13^C resonances at 93.1 ppm for **8a** and δ = 93.9/92.4 ppm for **9a** in accordance with the 93 ppm observed for the respective BTMSA complex [9]. The ^29^Si NMR down-field shift associated with the coordination to the Co_2_(CO)_6_ fragment amounts to ca 20 ppm for the alkynyl substituted silicon atom of the Si_2_Me_5_ units of **8a** and **9a** (Table 1) from δ = ca. −38 ppm for the free alkyne to ca −18 ppm for the complex. 

Complexes **2b** and **6b** are the dimolybdenum analogs of the cobalt complexes **2a** and **6a**. While for cobalt complex **2a** the ^13^C alkyne resonances were not much different from the free alkyne, the dimolybdenum compex **2b** features both of these resonances shifted down-field. The ^29^Si signal of the central silicon atom at −58.8 ppm (Table 1), however, indicates a marked deshielding, which is consistent with what has been found for propargyl cations and which is stronger than found in the dicobalt complex **2a** of the same alkyne (δ = −67.7 ppm). The situation for **6b** is quite similar. The ^13^C alkyne resonances at 93.1 and 108.1 ppm are very close to **2b** and also the ^29^Si chemical shift of the attached silicon atom at −16.8 ppm is again somewhat less shielded than in the corresponding dicobalt complex **6a**.

The NMR spectroscopic features of the dicobalt and dimolybdenum complexes of silylated alkynes show that the coordinated ligand resembles an alkane. In contrast to that, coordination to Ni(0) displays a different picture. The ^13^C NMR spectrum of complex **3c**, which features alkyne **1** as ligand exhibits resonances at 118.8 (C-Si) and 157.9 (C-Ph) ppm, which clearly indicates a more pronounced C-C double bond character of the coordinated alkyne. The ^29^Si NMR shift of the central hypersilyl silicon atom of **3c** (−90.0 ppm) is also in line with a vinyl substituted tris(trimethylsilyl)silane. ^13^C (113.5 and 160.4 ppm) and ^29^Si (−85.1) NMR shifts of (*E*)-*^t^*BuCH=CHSi(SiMe_3_)_3_ are supporting these assignments [38]. Very similar spectroscopic pictures were found for complex **2c**, which features coordinated alkyne **2**, and for **6c** with coordinated alkyne **6**. 

### 2.5. Crystallographic Analysis

In addition to the spectroscopic analysis of novel Co, Mo and Ni silylalkyne complexes, we have also studied a few of the new complexes by single crystal XRD analysis. Table 2 and Appendix A combine the obtained date with the previous examples from our earlier study [17].

As expected the germyl substituted alkyne complex compound **7a** (Figure 1) is isostructural to **1a** [17]. Both compounds crystallize in the monoclinic space group P2_1_. Structure parameters of **1a**, **7a**, and also **11a** such as Co-Co, C-C, and Co-C distances are very similar (see Table 2). A C-Ge bond for **7a** of 1.966(2) Å is quite typical and the spatial orientations of the phenyl and E(SiMe_3_)_3_ (E = Si, Ge) in **1a** and **7a** are almost identical and very similar to that of **11a**.

Compound **11a** (Figure 2) was found to crystallize in the monoclinic space group P2_1_/n with two crystallographically independent molecules in the asymmetric unit. In one of these molecules, the sulfur atom in the thienyl unit is disordered over two positions. The plane of the thienyl group is aligned with the Si-C-C-C plane of the former alkyne unit.

Since molybdenum is a second-row element the Mo-Mo distance in complex **2b** (Figure 3) of 2.966(1) Å is much longer than the 2.45–2.47 Å which are typical for the Co-Co distances in the dicobalthexacarbonyl complexes. Nevertheless, the C-C bond in **2b** of 1.337(10) Å is quite similar to what is found for dicobalt complexes. Direct comparison with complex **2a**, where the C-C distance is 1.310(15) Å reveals that the dimolybdenum complex is able to induce a higher degree of sp^3^ hybridization consistent with elongated C-C bond. This is also evident by a much-diminished C-C-Si angle of 132.1(7) deg for **2b** compared to the 142.7(8) deg found for **2a** [17]. 

In the course of the nickel complex synthesis we obtained crystals of our starting material (COD)Ni(dippe) [24] and determined its structure by single crystal XRD analysis (Figure 4). The complex crystallizes in the monoclinic space group C2/c and is quite similar to a number of other diphosphine nickel COD complexes [39,40,41,42,43]. Compared to similar complexes the P-Ni distances of 2.161(1) Å and the C-Ni distances between 2.094(2) and 2.107 Å are relatively short. Nevertheless, distances in the analogous dppe complex [44] are almost identical.

Complex **13c** (Figure 5) which crystallizes in the monoclinic space group P2_1_/c is the only nickel complex of this study that could be crystallographically analyzed. Its structural features are very close to that of the bis(trimethylsilyl)acetylene nickel complex with the dippe ligand [27] with Ni-P distances of 2.152(1) and 2.162(2) Å and a C-C distance of the coordinated alkyne of 1.296(8) Å.

## 3. Experimental Section

All reactions involving air-sensitive compounds were carried out under an atmosphere of dry nitrogen or argon using either Schlenk techniques or a glove box. Solvents were dried using a column solvent purification system [45]. If not, otherwise stated chemicals were obtained from different suppliers and were used without further purification. 

^1^H (300 MHz), ^13^C (75.4 MHz), ^29^Si (59.3 MHz), and ^31^P (124.4 MHz) NMR spectra were recorded on a Varian Unity INOVA 300 spectrometer. Samples for ^29^Si spectra were either dissolved in deuterated solvents or in cases of reaction samples measured with a D_2_O capillary in order to provide an external lock frequency signal. To compensate for the low isotopic abundance of ^29^Si the INEPT pulse sequence was used for the amplification of the signal [46,47]. If not noted otherwise the used solvent was C_6_D_6_ and all samples were measured at rt. Elementary analysis was carried using a Heraeus VARIO ELEMENTAR EL apparatus. 

For X-ray structure analyses, the crystals were mounted onto the tip of glass fibers, and data collection was performed with a BRUKER-AXS SMART APEX CCD diffractometer using graphite-monochromated Mo Kα radiation (0.71073 Å). The data were reduced to F^2^_o_ and corrected for absorption effects with SAINT [48] and SADABS [49,50], respectively. Structures were solved by direct methods and refined by full-matrix least-squares method (SHELXL97) [51]. All non-hydrogen atoms were refined with anisotropic displacement parameters. Hydrogen atoms were placed in calculated positions to correspond to standard bond lengths and angles. Crystallographic data (excluding structure factors) for the structures of compounds **7a**, **11a**, **2b**, **13c** and (COD)Ni(dippe) reported in this paper have been deposited with the Cambridge Crystallographic Data Center as supplementary publication no. CCDC-1881863 (**7a**)**,** 1881859 (**11a**), 1881860 (**2b**), 1881862 (**13c**), and 1881861 (COD)Ni(dippe). Copies of data can be obtained free of charge at: http://www.ccdc.cam.ac.uk/products/csd/request/. Figures of solid-state molecular structures were generated using Ortep-3 as implemented in WINGX [52] and rendered using POV-Ray 3.6 [53].

Co_2_(CO)_8_ was freshly sublimed before use. Cp_2_Mo_2_(CO)_4_ was obtained from Cp_2_Mo_2_(CO)_6_ by thermal treatment [22]. (COD)Ni(dippe) [24], tris(trimethylsilyl)silylphenylacetylene (**1**) [37], tris(trimethylsilyl)silylethyne (**2**) [54], trimethylsilylphenylacetylene (**6**) [55], tris(trimethylsilyl)germylphenylacetylene (**7**) [18], bis(pentamethyldisilanyl)ethyne (**8**) [19], 1-trimethylsilyl-2-pentamethyldisilanylethyne (**9**) [20], 1,4-bis[tris(trimethylsilyl)silyl]butadiyne (**10**) [18], tris(trimethylsilyl)germyl potassium [56], and 1,2-dichlorotetramethyldisilane [57,58] have been prepared according to literature procedures.

### 3.1. Synthesis of Oligosilanylalkynes

*1,2-Bis(pentamethydisilanyl)ethyne* (**8**) A solution of *n*BuLi (15.98 mmol, 2M in hexane) in Et_2_O (10 mL) and THF (10 mL) is cooled to −70 °C and trichloroethylene (5.33 mmol) in Et_2_O (10 mL) is added dropwise. The reaction mixture is allowed to warm up to r.t. and stirring is continued for 3 h. A white precipitate occurred. The reaction is cooled again to −70 °C and chlorpentamethyldisilane in Et_2_O (10 mL) is added dropwise. The stirring is continued for another 20 h at r.t. and then poured to an ice-cold mixture of 0.5 M H_2_SO_4_ und Et_2_O. The phases were separated, the organic layer washed with brine and then dried over Na_2_SO_4_. After filtration, the solvent was removed and subjected to Kugelrohr distillation (0.1 mbar, 50 °C) to separate the product from the also formed pentamethyldisilanylethyne (^29^Si: −18.8, −34.4.). Compound **8** was obtained as a colorless oil (0.404 g, 26%). NMR (δ in ppm): ^1^H: 0.19 (s, 12H), 0.14 (s, 18H). ^13^C: 115.3, −2.5, −2.8. ^29^Si: −19.4, −38.2.

*1-(Tris(trimethylsilyl)silyl)-2-(thien-2′-yl)ethyne* (**11**) A solution 2-bromothiophene (1960 mg, 12 mmol) in freshly distilled diisopropylamine (10 mL) was cooled to 0 °C, CuI (60 mg, 0.32 mmol), PPh_3_ (180 mg, 0.69 mmol), and Pd(OAc)_2_ (60 mg, 0.27 mmol) were added and the stirring continued for 30 min. After the addition of tris(trimethylsilyl)silylethyne (**2**) (4.2 mL, 30 mmol) the mixture was stirred for further 12 h at r.t. The solvent was removed and the residue treated with Et_2_O (50 mL) and 2M H_2_SO_4_. The layers were separated, the organic layer washed with saturated NaHCO_3_ solution and then dried with Na_2_SO_4_. The solvent was removed and the product objected to a flash chromatography on silica gel (eluents: pentane). Oily colorless **11** was obtained (1260 mg, 58%). NMR (δ in ppm, CDCl_3_): ^1^H: 7.16 (dd, *J* = 1 and 7.9 Hz, 1H), 7.15 (dd, *J* = 1.3 and 6.6 Hz, 1H), 6.93 (dd, *J* = 3.4 and 5.1 Hz, 1H), 0.27 (s, 27H, SiMe_3_). ^13^C: 131.3, 126.7, 126.1, 124.9, 100.4, 93.5, 0.4. ^29^Si: −11.4, −100.2. 

*2,5-Bis[tris(trimethylsilyl)silylethynyl)thiophene* (**12**) The reaction was done analogously to that for the preparation of **11** using 2,5-dibromothiophene (1230 mg, 5.08 mmol), freshly distilled diisopropylamine (10 mL), CuI (30 mg, 0.16 mmol), PPh_3_ (90 mg, 0.35 mmol), Pd(OAc)_2_ (30 mg, 0.14 mmol), and tris(trimethylsilyl)silylethyne (**2**) (2000 mg, 7.2 mmol). The mixture was stirred for further 12 h at 60 °C. Oily colorless **12** was obtained (2480 mg, 55%). NMR (δ in ppm, CDCl_3_): ^1^H: 7.29 (s, 2H), 0.68 (s, 27H, SiMe_3_). ^13^C: 131.4, 130.3, 129.6, 111.5, 0.4. ^29^Si: −11.3, −100.2. 

### 3.2. Oligosilanylalkyne Dicobalthexacarbonyl Complexes

*1-Phenyl-2-tris(trimethylsilyl)germylethyne-Co_2_(CO)_6_* (**7a**). A solution of tris(trimethylsilyl)germylphenylacetylene (**7**) (150 mg, 0.381 mmol) in pentane (2 mL) was added dropwise to Co_2_(CO)_8_ (130 mg, 0.381 mmol) dissolved in pentane (3 mL). The dark red solution was stirred for 1.5 h and then the solvent removed. Black crystalline **7a** (248 mg, 93%) was obtained. NMR (δ in ppm): ^1^H: 7.54 (m, 2H), 7.05 (m, 2H), 6.95 (m, 1H), 0.31 (s, 27H, SiMe_3_). ^13^C: 201.2, 201.1, 139.4, 130.0, 128.8, 128.0, 109.7, 80.9, 2.7. ^29^Si: −4.4. Anal. Calcd. For C_23_H_32_Co_2_GeO_6_Si_3_ (679.23): C 40.67, H 4.75. Found: C 40.49, H 4.66.

*1,2-Bis(pentamethyldisilanyl)ethyne-Co_2_(CO)_6_* (**8a**). The reaction was done analogously to that for the preparation of **7a** using Co_2_(CO)_8_ (83 mg, 0.244 mmol) and bis(pentamethyldisilanyl)ethyne (**8**) (70 mg, 0.244 mmol). A dark red oil of **8a** (113 mg, 81%) was obtained. NMR (δ in ppm): ^1^H: 0.37 (s, 12H), 0.15 (s, 18H). ^13^C: 201.8, 196.0, 93.1, 0.3, −1.6. ^29^Si: −15.9, −18.3.

*1-Trimethylsilyl-2-pentamethyldisilanylethyne-Co_2_(CO)_6_* (**9a**). The reaction was done analogously to that for the preparation of **7a** using Co_2_(CO)_8_ (244 mg, 0.563 mmol) and 1-trimethylsilyl-2-pentamethyldisilanylethyne (**9**) (150 mg, 0.563 mmol). Black crystalline **9a** (324 mg, 95%) was obtained. NMR (δ in ppm): ^1^H: 0.33 (s, 6H), 0.26 (s, 9H), 0.12 (s, 9H). ^13^C: 201.2, 93.9, 92.4, 1.3, −0.2, −1.9. ^29^Si: 0.3, −15.8, −18.6.

*1,4-Bis[tris(trimethylsilyl)silyl]buta-1,3-diyne-Co_2_(CO)_6_* (**10a**). Reaction was done according to **7a** using Co_2_(CO)_8_ (63 mg, 0.184 mmol) and **10** (100 mg, 0.184 mmol). After cooling the solution to −70 °C black crystalline **10a** (136 mg, 89%) was obtained. NMR (δ in ppm): ^1^H: 0.88 (s, 27H), 0.79 (s, 27H). ^13^C: 201.1; 105.1; 103.9; 88.6; 72.6; 2.1; 0.5. ^29^Si: −10.6; −10.9; −65.9; −99.2.

*1-(2′-Thienyl)-2-tris(trimethylsilyl)silylethyne-Co_2_(CO)_6_* (**11a**). A solution of trimethylsilyl-2′-thienylacetylene (**11**) (200 mg, 0.564 mmol) in pentane (2 mL) was added dropwise to Co_2_(CO)_8_ (193 mg, 0.564 mmol) dissolved in pentane (3 mL). The dark red solution was stirred for 3 h and then cooled to −70 °C. Black crystalline **11a** (296 mg, 82%) was obtained. NMR (δ in ppm): ^1^H: 7.08 (dd, *J* = 3.6 and 1.3 Hz), 6.68 (dd, *J* = 5.2 and 1.2 Hz), 6.54 (dd, *J* = 5.2 and 3.6 Hz), 0.32 (s, 27H). ^13^C: 200.5, 200.4, 200.3, 143.2, 128.0, 127.8, 126.3, 96.6, 76.9, 2.3. ^29^Si: −11.9, −67.1.

*2,5-Bis[tris(trimethylsilyl)silylethynyl]thiophene-Co_2_(CO)_6_* (**12a**). The reaction was done analogously to that for the preparation of **7a** using Co_2_(CO)_8_ (164 mg, 0.480 mmol) and 2,5-bis[tris(trimethylsilyl)silylethynyl)thiophene (**12**) (150 mg, 0.240 mmol). After cooling the solution to −70 °C black crystalline **12a** (218 mg, 76%) was obtained. NMR (δ in ppm): ^1^H: 6.23 (s, 2H), 0.28 (s, 54H). ^13^C: 201.3, 130.7, 127.9, 112.2, 111.8, 2.3. ^29^Si: −11.9, −66.9.

### 3.3. Oligosilanylalkyne Dimolybdenyum-1,2-dicyclopentadienyltetraacarbonyl Complexes

*Tris(trimethylsilyl)silylethyne-Cp_2_(CO)_4_Mo_2_* (**2b**). Cp_2_Mo_2_(CO)_6_ (719 mg, 1.47 mmol) in diglyme (10 mL) was kept under reflux for 3h. After cooling to rt, tris(trimethylsilyl)silylethyne (**2**) (200 mg, 0.733 mmol) in THF (3 mL) was added. After 14 d the solvent was removed, the residue treated with pentane and the insoluble remaining removed by centrifugation. Pentane was removed and **2b** (270 mg, 52%) was obtained as a red oily residue obtained which crystallized after several days. NMR (δ in ppm): ^1^H: 6.46 (s, 1H), 4.98 (s, 10H, *Cp*), 0.26 (s, 27H, Si*Me*_3_). ^13^C: 241.2 (*C*O), 233.0 (*C*O), 101.9, 96.6, 92.4 (*Cp*), 3.4. ^29^Si: −11.8, −58.8.

*1-Phenylethynylpentamethyldisilan-Cp_2_(CO)_4_Mo_2_* (**6b**). The reaction was done analogously to that for the preparation of **2b** using Cp_2_Mo_2_(CO)_6_ (843 mg, 1.72 mmol) and pentamethyldisilanylphenylacetylene (**6**) (200 mg, 0.860 mmol). Recrystallization with pentane gave red crystalline **6b** (320 mg, 56%). NMR (δ in ppm): ^1^H: 7.55 (m, 2H), 7.25 (m, 2H), 7.02 (m, 1H), 5.03 (s, 10H, *Cp*), 0.55 (s, 6H, Si*Me*_2_), 0.28 (s, 9H, −Si*Me*_3_). ^13^C: 236.8, 231.8, 231.5, 147.2, 130.0, 128.2, 126.5, 116.0, 92.1, 1.4, 0.0. ^29^Si: −7.1, *−16.8*.

### 3.4. Oligosilanylalkyne Nickel-Ethylenebis(diisopropylphosphine) Complexes

*Di-iso-propylphosphinoethylene[tris(trimethylsilyl)silylethynyl]nickel* (**2c**). (COD)Ni(dippe) (56 mg, 0.130 mmol) and tris(trimethylsilyl)silylethyne (**2**) (36 mg, 0.130 mmol) were dissolved in toluene (3 mL) and stirred for 4 d at 65 °C. The progress of the reaction was checked by NMR. After removing the solvent dark green crystalline **2c** (59 mg, 85%) was obtained. NMR (δ in ppm): ^1^H: 8.12 (dd, *J*_H-P_ = 11.2 Hz, *J*_H-P_ = 31.8 Hz, 1H), 2.01 (m, 2H), 1.90 (m, 2H), 1.29–1.15 (m, 10H), 1.02–0.81 (m, 18H), 0.39 (d, 27H, *J*_H-P_ = 3 Hz, SiMe_3_). ^13^C: 145.3 (dd, *J*_C-P_ = 8 Hz, *J*_C-P_ = 34 Hz), 124.3 (dd, *J*_C-P_ = 7 Hz, *J*_C-P_ = 42 Hz), 25.7, 19.6 (d, *J*_C-P_ = 7 Hz), 18.9, 1.6 (SiMe_3_). ^29^Si: −14.0 (d, *J*_Si-P_ = 3 Hz, SiMe_3_), −89.4 (dd, *J*_Si-P_ = 2 Hz, *J*_Si-P_ = 15 Hz, Si_q_). ^31^P: 81.6 (d, *J*_P-P_ = 48.5 Hz), 76.0 (d, *J*_P-P_ =48.5 Hz).

*Di-iso-propylphosphinoethylene-[1-phenyl-2-tris(trimethylsilyl)silylethynyl]nickel* (**3c**). (COD)Ni(dippe) (130 mg, 0.303 mmol) and tris(trimethylsilyl)silylphenylacetylene (**1**) (106 mg, 0.303 mmol) were dissolved in toluene (4 mL) and stirred for 7 d at 65 °C. The progress of the reaction was checked by NMR. After removing the solvent dark green crystalline **3c** (166 mg, 82%) was obtained. NMR (δ in ppm): ^1^H: 7.23 (m, 1H), 7.15 (m, 2H), 6.94 (m, 2H), 2.10 (m, 2H), 1.62 (m, 2H), 1.21 (m, 4H), 0.98 (m, 24H), 0.34 (s, 27H). ^13^C: 157.9 (dd, *J*_C-P_ = 5 Hz, *J*_C-P_ = 32 Hz), 146.4 (dd, *J*_C-P_ = 4 Hz, *J*_C-P_ = 14 Hz), 128.4, 127.9, 126.4 (d, *J*_C-P_ = 2 Hz), 123.8 (d, *J*_C-P_ = 2 Hz), 118.6 (dd, *J*_C-P_ = 6 Hz, *J*_C-P_ = 42 Hz), 26.1 (dd, *J*_C-P_ = 5 Hz, *J*_C-P_ =13 Hz), 24.9 (dd, *J*_C-P_ = 4 Hz, *J*_C-P_ = 16 Hz), 22.0 (d, *J*_C-P_ = 11 Hz), 19.8 (d, *J*_C-P_ = 8 Hz), 18.7 (d, *J*_C-P_ = 16 Hz), 2.2. ^29^Si: −14.1 (d, *J*_Si-P_ = 3 Hz), −90.0 (t, *J*_Si-P_ = 14 Hz).^31^P: 78.3 (d, *J*_P-P_ = 49.1 Hz), 77.2 (d, *J*_P-P_ = 49.1 Hz).

*Di-iso-propylphosphinoethylene(1-phenyl-2-pentamethyldisilanylethynyl)nickel* (**6c**). (COD)Ni(dippe) (82 mg, 0.191 mmol) and pentamethyldisilanylphenylacetylene (**6**) (44 mg, 0.191 mmol) were dissolved in toluene (3 mL) and stirred for 18 h at 65 °C. The progress of the reaction was checked by NMR. After removing the solvent dark green oily **6c** (92 mg, 87%) was obtained. NMR (δ in ppm): ^1^H: 7.44 (d, 2H), 7.21 (t, 2H), 7.02 (t, 1H), 1.99 (sept, 2H), 1.80 (m, 2H), 1.24-1.16 (m, 10H), 0.97-0.79 (m, 18H), 0.55 (s, 6H, SiMe_2_), 0.19 (s, 9H, SiMe_3_). ^13^C: 159.6 (dd, *J*_C-P_ = 5 Hz, *J*_C-P_ = 34 Hz), 144.8 (dd, *J*_C-P_ = 5 Hz, *J*_C-P_ = 12.1 Hz), 132.2 (dd, *J*_C-P_ = 4 Hz, *J*_C-P_ = 32 Hz), 128.8, 126.7 (d, *J*_C-P_ = 3 Hz), 124.2 (d, *J*_C-P_ = 1 Hz), 26.2 (dd, *J*_C-P_ =5 Hz, *J*_C-P_ = 14 Hz), 25.5 (d, *J*_C-P_ = 5 Hz, *J*_C-P_ = 16 Hz), 22.0 (t, *J*_C-P_ = 20 Hz), 21.8 (t, *J*_C-P_ = 18 Hz), 20.8 (d, *J*_C-P_ = 9 Hz), 19.8 (d, *J*_C-P_ = 8 Hz), 19.0 (d, *J*_C-P_ = 21 Hz), −0.2 (d, *J*_C-P_ = 2 Hz, SiMe_2_), −1.2 (SiMe_3_). ^29^Si: −19.6 (d, *J*_Si-P_ = 4 Hz, SiMe_3_), −31.4 (dd, *J*_Si-P_ = 4 Hz, *J*_Si-P_ = 14 Hz, SiMe_2_). ^31^P: 78.3 (d, *J*_P-P_ = 49.0 Hz), 77.3 (d, *J*_P-P_ = 49.0 Hz).

*Di-iso-propylphosphinoethylene(1-(2′-thienyl)-2-tris(trimethylsilyl)silylethynyl)nickel* (**13c**). Reaction was done according to **2c** using (COD)Ni(dippe) (0.172 mmol) and **13** (61 mg, 0.172 mmol). Orange crystals of **13c** (72 mg, 75%) were obtained. NMR (δ in ppm): ^1^H: 7.01 (dd, *J* = 3.5 and 1.3 Hz), 6.76 (dd, *J* = 5.2 and 1.2 Hz), 6.40 (dd, *J* = 5.2 and 3.5 Hz), 2.14 (m, 2H), 1.63 (m, 2H), 1.14 (m, 4H), 1.01 (m, 24H), 0.17 (s, 9H). ^13^C: 154.2 (dd, *J*_C-P_ = 5 Hz, *J*_C-P_ = 30 Hz), 148.3 (dd, *J*_C-P_ = 4 Hz, *J*_C-P_ = 14 Hz), 143.2 (dd, *J*_C-P_ = 5 Hz, *J*_C-P_ = 39 Hz), 129.3, 126.6, 126.1, 25.6 (dd, *J*_C-P_ = 5 Hz, *J*_C-P_ =13 Hz), 24.9 (dd, *J*_C-P_ = 4 Hz, *J*_C-P_ = 12 Hz), 22.5 (d, *J*_C-P_ = 12 Hz), 19.7 (d, *J*_C-P_ = 8 Hz), 19.2 (d, *J*_C-P_ = 16 Hz), 5.2. ^29^Si: −19.1 (d, *J*_Si-P_ = 12 Hz).^31^P: 79.1 (d, *J*_P-P_ = 51 Hz), 77.7 (d, *J*_P-P_ = 51 Hz).

## 4. Conclusions

Reactivity enhancement of metal coordinated halo- and hydrosilylalkynes is a well-documented fact [10,11,14,15,16]. In a previous study, we have shown that oligosilanylalkyne coordination to the dicobalthexacarbonyl fragment results in the formation of highly reactive compounds [17]. This reactivity was explained as activation of Si-Si bonds in accordance to what was observed previously for Si-H bonds. ^29^Si NMR spectroscopic analysis confirmed down-field shift of the alkynyl substituted silicon atom resonances, which is in agreement with increased electrophilic properties. The current study extends the range of oligosilanylalkyne complexes to a number of new dicobalthexacarbonyl complexes but also to 1,2-bis(cyclopentadienyl)tetracarbonyldimolybdenum and (dippe)Ni complexes. Similar to what was observed before for metal stabilized propargyl cations, also for the oligosilanylalkyne complexes the respective dimolybdenum complexes were found to cause even stronger ^29^Si NMR deshielding than was found for the dicobalt complexes. While the dimetallic complexes cause rehybridization of the alkyne carbon atoms to sp^3^ (alkane type), in the respective nickel complexes one of the π-bonds of the alkynes was found to be retained. The NMR spectroscopic properties of the nickel complexes are completely different from those of the dicobalt and dimolybdenum complexes. In the latter, strongly deshielded ^29^Si NMR chemical shifts of the attached silicon atoms indicate enhanced reactivity, whereas the ^29^Si NMR resonances of the respective nickel complexes are similar to that of respective vinylsilanes. Thus, the strong deshielding of the alkyne attached silicon atoms was not observed for the nickel complexes and therefore no pronounced Si-Si bond activation can be expected.

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
