# Peer review of "Spectroscopic and Structural Study of Some Oligosilanylalkyne Complexes of Cobalt, Molybdenum and Nickel"

_molecules, 2019, doi:10.3390/molecules24010205_

Reviewer 1 Report

Introduction is well written, and very interesting. Further, the characterization of novel products is sound, and I recommend publication in Molecules.

The author used Pd(Ac)2 in the experimental section for the synthesis of alkynylsilane. If Pd(Ac)2 indicates palladium acetate, the abbreviation “Pd(OAc)2” is more general.

Author Response

The reviewer is completely right. This should be Pd(OAc)2

Reviewer 2 Report

The manuscript submitted by Marcschner, Baumgartner and co-workers continues the progress presented in Organometallics 2006, 25, 4897–4908 and showcases the synthesis of metal complexed oligosilanylated alkynes using Co, Mo, and Ni.

While I think the reactivity presented in the manuscript is interesting I find reading the paper a bit difficult. For instance the introduction could benefit from more organization. It is not very clear what the previous work is and what the current goals are. I strongly suggest including a Scheme with some of the previous results which also highlights some of the results from this manuscript.

Secondly, this manuscript does not have a Conclusions section. I think this part is very important for the reader to understand the most important aspects of this work and also what the future directions are. Oddly, the 3rd section which is also called “Discussion” seems to serve more like a short summary of this paper, but it does not showcase the importance of this work…

The Results and discussion part is very rich in organometallic synthesis, but it is hard to discern which of the complexes are new and which complexes were already published. I understand that the authors want to present the current results by comparing them with the previous ones, but the current set up makes this read difficult. This part is not linked well with the Introduction and I find organizing this section by the metal to be hard to read and understand. For instance, why did the authors decide to use substrate 12? I just feel this section includes a list of experiments, but which are not properly explained.

Otherwise, the compounds are well characterized. If the compound was already reported, then it should be stated and have the reference given. However, except for compound 7a, I don’t see any other elemental analysis data. Is there a reason for this? Also, organic compounds should have HR-MS data included as well.

On a side, this text includes a number of typos. I did not find all of them, but could spot some at line 46: “dicobaltbut” and at line 50 “proofed” should be switched with “proved”.

Provided that the above points are addressed, I recommend acceptance of this manuscript in Molecules.

Author Response

In order to clarify the main intention of the paper we have changed the title to "Spectroscopic and Structural Study of Some Oligosilanylalkyne Complexes of Cobalt, Molybdenum and Nickel". This is a much better description of the paper.

We have added another scheme to the introduction to clarify the metal induced reactivity enhancement. At the end of the introduction, it is also mentioned that the current study aims at extending the spectroscopic and structural studies to more elaborate oligosilanyl alkynes and to other transition metals.

The reviewer is correct that we mistitled Section 3 as “Discussion” while it actually is a summary and conclusion.

We have added a sentence outlining as to why we have chosen to introduce thienyl groups.

Unfortunately, HR-MS is a difficult issue at our department and we do not get this on a routine basis. We prefer not to comment on this in the paper.

Reviewer 3 Report

This manuscript describes the preparations and characterizations of a series of Nickel, Cobalt and Molybdenum complexes having oligosilanylalkyne derivatives as ligands.  The formations of the resulting complexes are not unusual, while somewhat interesting.  The experimental works are through and high quality.  I think the results should be of general interest to organometallic chemists.  Thus, the report is in my opinion suitable for publication in molecules.  However, there are a number of errors in English that should be edited out before the paper is accepted.  The English of the paper should be checked, corrected and improved by a native speaker of English.

Author Response

We have given the paper another check for issues with the English.